# Modeling the Trajectories of Ballistics in the Summit Area of Mt. Etna (Italy) during the 2020–2022 Sequence of Lava Fountains

Giorgio Costa [1], Luigi Mereu [2], Michele Prestifilippo [3], Simona Scollo [3,*] and Marco Viccaro [1,3]

1 Dipartimento di Scienze Biologiche, Geologiche e Ambientali, Università degli Studi di Catania, 95129 Catania, Italy
2 Sezione di Bologna, Istituto Nazionale di Geofisica e Vulcanologia, 40127 Bologna, Italy
3 Istituto Nazionale di Geofisica e Vulcanologia, Osservatorio Etneo, 95125 Catania, Italy
* Correspondence: simona.scollo@ingv.it

**Abstract:** Between 2020 and 2022, more than sixty lava fountains occurred at Mt. Etna (Italy), which formed high eruption columns rising up to 15 km above sea level (a.s.l.). During those events, several ballistics fell around the summit craters, sometimes reaching touristic areas. The rather frequent activity poses questions on how the impact associated with the fallout of those particles, can be estimated. In this work, we present field data collected soon after the lava fountain on 21 February 2022. This event produced a volcanic plume of about 10 km a.s.l. which was directed toward the southeast. Several ballistics fell in the area of the Barbagallo Craters (just southeast of the summit area at around 2900 m a.s.l.), which is one of the most popular touristic areas on Etna. Hence, we collected several samples and performed laboratory analyses in order to retrieve their size, shape and density. Those values together with a quantitative analysis of the lava fountain were compared with results obtained by a free-available calculator of ballistic trajectories named the 'Eject!'. A similar approach was hence applied to other lava fountains of the 2020–2022 sequence for which the fallout of large clasts was reported. This work is a first step to identifying in near real-time the area affected by the fallout of ballistics during Etna lava fountains and quantifying their hazard.

**Keywords:** Mt. Etna (Italy); lava fountains; fallout of ballistics; Eject! sofware; hazard from ballistics

## 1. Introduction

Basaltic systems are particularly acknowledged for producing predominantly effusive activity, due to the physical and chemical properties of the magmas and eruptive dynamics. However, even basaltic volcanoes can produce explosive activity and, at Etna, it is mainly characterized by the formation of lava fountains, the height of which primarily depends on the content of dissolved volatiles in the magma, the eruptive rate and the geometry of conduit/vent [1]. In recent decades, Etna has given rise to several sequences of lava fountains, also known locally as paroxysms, as occurred in 1998–1999, 2000, 2007–2008, 2011–2013, and 2015–2016 [2–9]. The last sequence began on 13 December 2020, and ended on 21 February 2022, producing sixty-two paroxysmal episodes from the South-East Crater (SEC).

Lava fountains develop following a precise evolution, through a progressive intensification of the explosive activity that, from weakly Strombolian, culminates rapidly toward a fountaining phase. The most intense phase of this kind of activity produces sustained lava jets [10] on average 500 m high but for some episodes reaching even 1500 m in height (e.g., 23 February 2021 and 10 February 2022). Moreover, paroxysmal episodes produce dense eruptive columns of tephra, sometimes as high as 15 km a.s.l. [11]. The duration of the entire eruptive phenomenon is about 2–3 h, while the intense fountaining is on the order of about 15–60 min. This activity also forms lava flows and leads to the fallout of

several ballistics which can be ejected up to distances of kilometers from the summit craters, thus constituting a potential threat for the volcano visitors, especially when this type of activity evolves very rapidly [12]. Due to the high occurrence of these types of events, the estimation of the impact near the most frequented areas of the volcano is necessary. Indeed, the fallout of larger clasts (>5 cm in size) can be a significant threat for people who annually visit Etna, but also for vehicles and infrastructure located on the lower slopes of the volcano, which have suffered repeated damages during the most energetic events [12,13]. Specifically, within a radius of 10 km of the Regional Etna Natural Park, there are ~1390 km of trails, paved roads, ~4600 buildings including commercial and residential properties and the touristic hubs of Rifugio Sapienza and Piano Provenzana with their Etna Cableway and ski installations [14].

In the present study, data collected in the field just after the 21 February 2022 lava fountain (Figure 1a) episode are presented. Several ballistics were found in the Barbagallo Craters area (between about 2700 and 2900 m a.s.l.), which is one of the most popular tourist areas on the southern side of Etna (Figure 1b), about 1.5 km from the SEC. This eruption produced an eruptive column of about 10 km (Figure 2A) in height and dispersion of pyroclastic material toward the southeast (Figure 2B). The collected samples have been analyzed for deriving their main physical features, such as size, shape, density and drag coefficient. These data were also used to estimate, through a free available software, the ballistic trajectories and distance reached for the eruption analyzed and for other paroxysmal events that occurred in 2021. The present study is, therefore, aimed at finding the possible impact areas of ballistics and the fallout distances reached during paroxysmal activities of Etna in order to better evaluate the hazard associated with these phenomena.

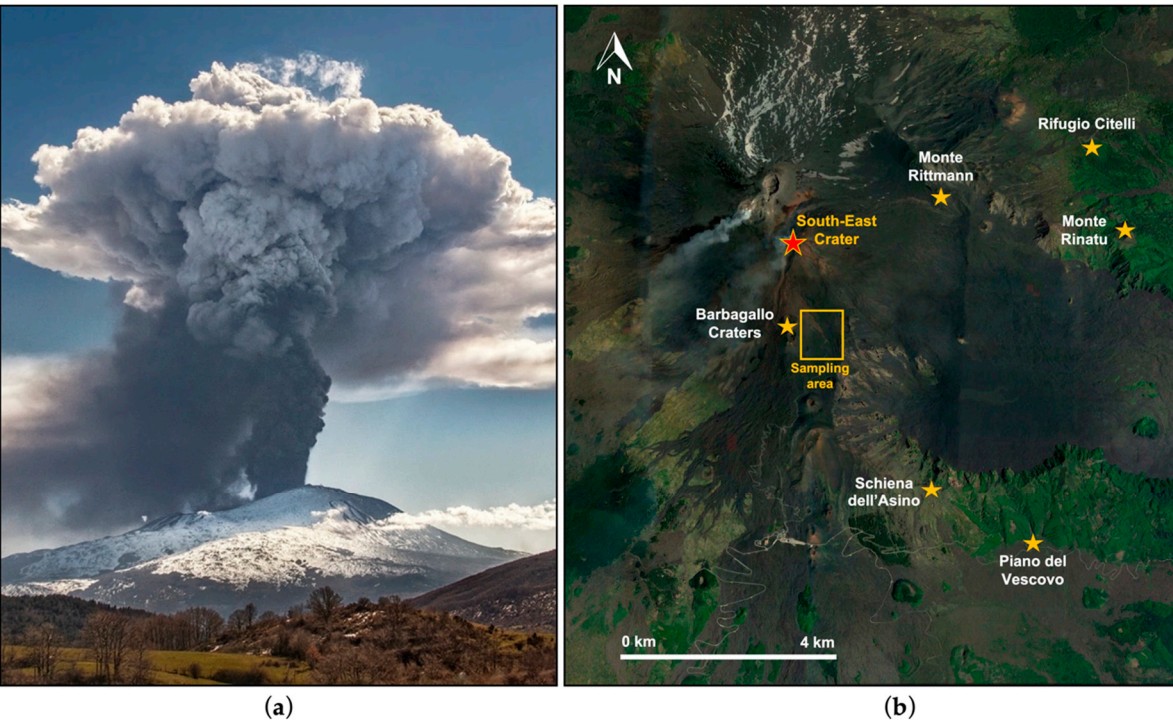

(a)    (b)

**Figure 1.** (**a**) The eruptive column produced by the February 21 paroxysmal episode in 2022, as seen from the northern slope of the volcano (photo by G. Costa); (**b**) Map of Etna's summit area taken from Google Earth, showing the places of interest in the present study. The orange rectangle shows the sampling area. The red star shows the position of the South-East Crater (SEC), which produced the paroxysmal events described in the present study. The orange stars indicate some tourist locations affected by fallout of large clasts during other paroxysmal episodes modeled in this study.

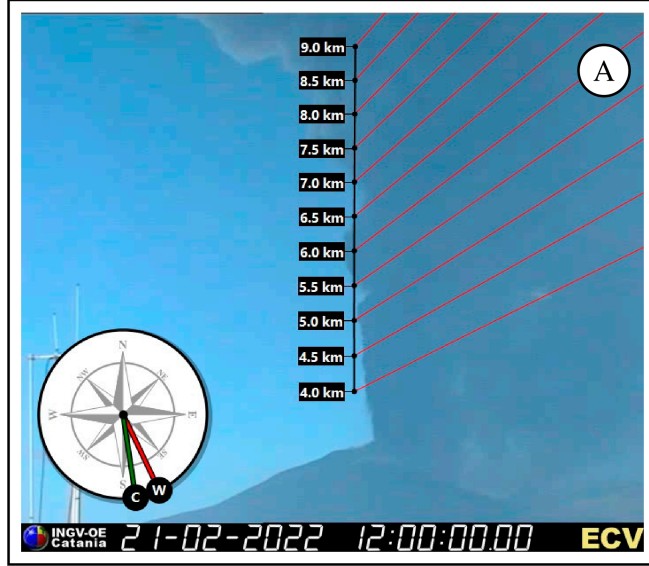
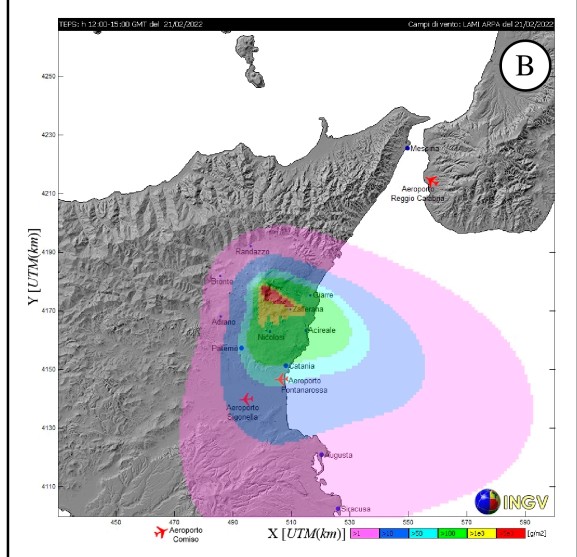

**Figure 2.** (**A**) Eruption column height estimated by the visible calibrated camera ECV of the Istituto Nazionale di Geofisica e Vulcanologia, Osservatorio Etneo (INGV-OE) [11,15]; (**B**) simulation of tephra fallout deposit used in the monitoring and forecasting system of INGV-OE [16].

## 2. Materials and Methods

### 2.1. Field Data Collection

Sampling was carried out on the southern side of Etna summit craters at an elevation between approximately 2600 and 2800 m a.s.l. (Figure 3), in order to analyze the deposit of ballistics produced by the paroxysmal activity that occurred on the morning of 21 February 2022 (Figure 1), as confirmed by field surveys carried out before and after the eruptive event.

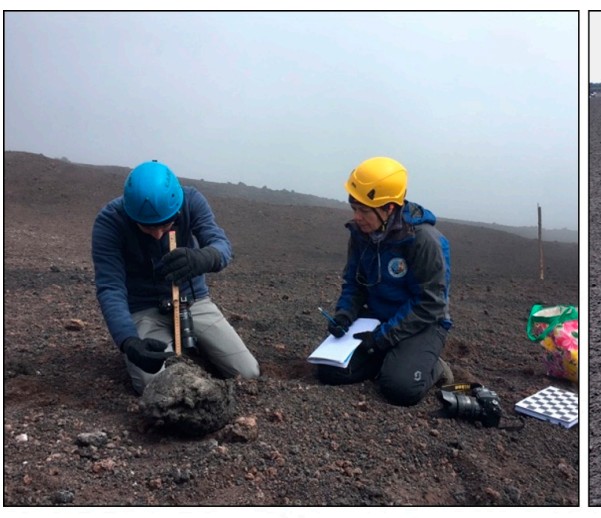
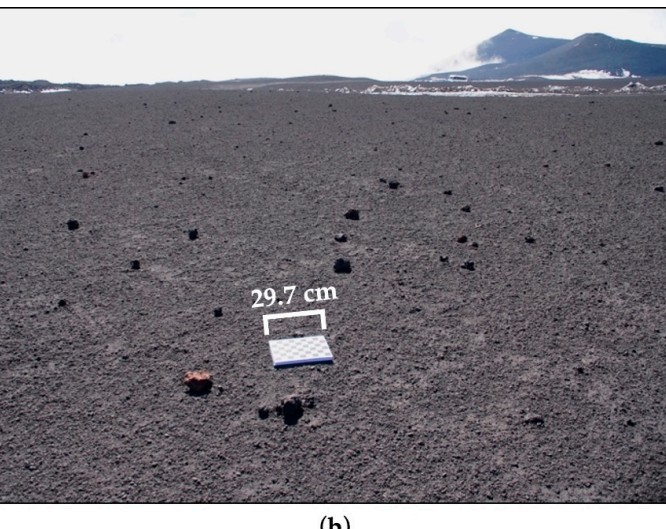

(**a**)  (**b**)

**Figure 3.** The sampling campaign was carried out on the southern side of Etna, at an altitude between 2600 and 2800 m a.s.l. (**a**) Measurements of one of the principal axes of a decimetric bomb. (**b**) Ballistics found on the ground level at about 2700 m.

Most of the collected clasts range from centimeters to decimeters in diameter. Some of those clasts were collected and analyzed in the laboratory. We measure their weight with a digital balance (in kilograms, kg). The three main axes (a, b, c), i.e., length, width and thickness (m) along the three perpendicular directions were measured using a pycnometer and/or meter, both in the field and laboratory. Hence, we estimated the aspect ratio (AR), defined as the ratio (c/a), which reflects the overall elongation of the clast [17] and the

geometrical diameter, defined as the cube root of the product of a, b, c. The density (kg/m$^3$) of five clasts was measured in the laboratory, measuring the water displacement of five clasts in a graduated cylinder.

### 2.2. Analysis of Images from the INGV Thermal Camera

Lava fountains can be approximated using saturated regions of the thermal cameras and, for this reason, have been named the "incandescent jet region" (IJR, hereafter) [10]. The characterization of the height of the IJR region produced during the paroxysmal episodes was carried out by analyzing images recorded by the thermal infrared (TIR) camera, located about 15 km south of the summit craters of Mt. Etna (ENT, Nicolosi), which is part of the Istituto Nazionale di Geofisica e Vulcanologia, Osservatorio Etneo (INGV-OE) permanent monitoring system.

The TIR camera provides a time series of 640 × 480 pixel images with a spatial resolution of several meters [18,19], and a thermal time series of 640 × 480 pixel images with a thermal sensitivity of 8.0 × 10$^{-3}$ K at 25 °C [10]. Images are displayed with a fixed color scale with a range from −20 °C to 70 °C, in order to assess the maximum height of the IJR by selecting the saturation portion of the images. This portion depends on the intrinsic properties of the camera and atmospheric factors, which can vary depending on the distance between the lava fountain and the camera itself. Most of the procedures used to identify the height of the IJR are based on setting a threshold to identify the zone of maximum saturation.

If we consider the IJR height as a proxy of the height of the lava fountain, it can then be related to an exit velocity $v_{ex}$ through the well-known Torricelli equation for a non-viscous ballistic flow [10,19–21]. However, in order to apply this equation, changes in atmospheric density and drag forces are considered negligible [10]. This methodology has been extensively applied at Etna [10,22,23].

Based on these assumptions, we can calculate the estimated IJR and relative vertical outflow velocity from the active vent (and vice versa), at each instant of time (*t*), through the following relationships:

$$v_{ex}(t) = \sqrt{2gH_{IJR}(t)} \tag{1}$$

Knowing the cross-section of the conduit, it is then possible to estimate, as a first approximation, the flow of gas and pyroclastic material, here named mass eruption rate (MER), escaping per unit time through the conduit [19,24]. Specifically, $v_{ex}$ (m/s) is the velocity of the gas and pyroclastic mixture, *g* (m/s$^2$) is the acceleration of gravity, and H$_{IJR}$ is the height of the IJR, expressed in meters above the crater rim. The conduit section of the SEC was considered circular and with a vent diameter of about 30 m [19,23,25]. After estimating the H$_{IJR}$ in pixels, the ENT camera was also calibrated in order to convert the height of the IJR from pixels to meters knowing the position of the thermal camera and the coordinates of the summit craters [22].

### 2.3. The Eject! Software

A free trajectory calculator for ballistic fragments ejected during explosive eruptions, available to the volcanological community through the portal https://vhub.org/ and https://pubs.er.usgs.gov/publication/ofr0145 (accessed on 28 December 2022), was used for modeling the particle trajectory. The software is called "Eject!" [26] and was written in Microsoft Visual Basic$^®$ and operates on any personal computer running Microsoft®Windows 95 or later allowing calculation of the trajectory of bombs/blocks, the maximum distance reached from the point of emission, the final fallout velocity and the travel time. The Eject! program requires the following input data: the initial ejection velocity (m/s), the ejection angle in degrees from horizontal, the vertical distance between the takeoff point and the landing point (m), particle information (in terms of density and diameter) and the air drag coefficient.

The acceleration of an individual particle can be calculated using standard, two-dimensional equations of motion:

$$\frac{dv_x}{dt} = \frac{F_x}{m} = \frac{-(v_x - w_x)\,\rho_a ||v - w||\,A\,C_d}{2m} \tag{2}$$

$$\frac{dv_z}{dt} = \frac{F_z}{m} = \frac{-(v_z - w_z)\,\rho_a\,||v - w||\,A\,C_d}{2m} - g\frac{\rho_r - \rho_a}{\rho_r} \tag{3}$$

where $dv_x/dt$ (m/s$^2$) is the acceleration of the clast in the horizontal direction (x), $dv_z/dt$ (m/s$^2$) is the acceleration of the clast in the vertical direction (z); $F_x$ and $F_z$ [(kg m)/s$^2$] are the forces along the x direction and along the z direction, respectively; $m$ (kg) is the mass of the clast; $v$ (m/s) is the velocity of the clast; w (m/s) is the wind velocity; $\rho_a$ (kg/m$^3$) is the density of the surrounding air; $\rho_r$ (kg/m$^3$) is the density of the clast; $A$ (m$^2$) is the frontal area of the clast; $C_d$ is the drag coefficient; $t$ (s) is the time; $g$ is the acceleration of gravity (9.81 m/s$^2$).

The software also includes atmospheric properties, such as: wind speed (m/s), sea level temperature (C$^\circ$), thermal gradient (C$^\circ$/km) and the elevation of the takeoff point above sea level (m), which is the elevation of the particle emission point. The atmospheric properties can be obtained from weather data forecasts [16] and are used by the program to calculate the air density, pressure and speed of sound as a function of altitude during the trajectory of the erupted clast. The data used in this paper were obtained from the spatial interpolation of the vertical profiles of the meteorological data provided by the HydroMeteorological Service of the Emilia-Romagna Regional Agency for Environmental Protection (ARPA-SIM), in northern Italy. These provide GRIB (GRIdded Binary) files packed in a binary format to increase storage efficiency. ARPASIM GRIB files are produced using the Cosmo model and are provided every 12 h with a time step of 3 h and the weather forecasts are given until 72 h. The ARPASIM grid covers an area rotated with respect to the Equator that is moved to the medium latitudes. It spans from 11.02° to 19.50° E and from 33.96° to 41.02° N and has 14 isobaric levels. The GRIB files are formed from 141 × 166 points stepped by 0.045°.

Other input information is related to the clast and includes its density (kg/m$^3$), diameter, or area. Specifically, the geometrical diameter of the three directions perpendicular to the clast (a, b, c) was considered in the present study. Different types of clast shapes, denoted as spheres, cubes and artillery shells can be considered to estimate the variable drag coefficient throughout the trajectory. We mark that very small blocks are also considered in our study which could be subject to other influences (e.g., jet updrafts, interactions with other blocks, broken during the fallout).

The last input properties are those related to the drag force, to which the block is subjected during its trajectory, and include the drag coefficient, which is a dimensionless coefficient varying depending on the shape of the block as a function of two parameters, i.e., the Reynolds number (*Re*) and the Mach number (*M*). Considering the size range of the ballistics measured in the field, the Reynolds numbers range between about $1 \times 10^5$ and $4.5 \times 10^5$ whereas the Mach number is between 0.1 and 0.2.

## 3. Results

### 3.1. Sampling Results

The fallout deposit produced during the paroxysmal episode of 21 February 2022 investigated on the southern side of Etna, was characterized by centimeter-sized lapilli (mostly <5 cm) scattered in a continuous blanket, above which there was the presence of scoria varying in size from about 4 to 12 cm, mixed with scattered, decimeter-size bombs (considered in our analysis) (Figure 4) (Table 1). Clasts collected in the field were mostly in the lower limit of blocks (>64 mm). We also found large lapilli of about 5 cm size (Figure 4d and Table 1).

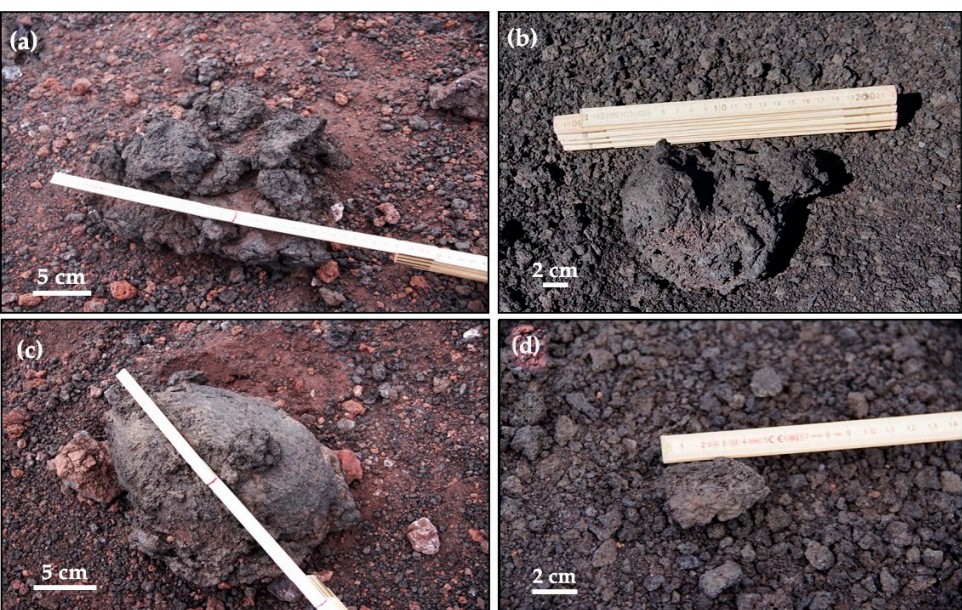

**Figure 4.** The pyroclastic material with different grain-size produced by the lava fountaining of 21 February 2022 and collected in the field (**a,b**). Continuous fall deposit of small lapilli scattered on the ground level, above which centimeter-sized clasts were found (**c,d**). Decimetric bombs found on the ground.

**Table 1.** Small blocks and large lapilli obtained from the analysis of all the samples taken for this study. LAT = Latitude; LON: Longitude; ALT: Altitude; D= distance from the crater; a, b and c are length, width and thickness along the three perpendicular directions, dg is the geometrical diameter obtained by the cube root of the product of a, b and c, M is the mass and AR is the aspect ratio.

| Ballistics | LAT | LON | ALT (m) | D (m) | a (m) | b (m) | c (m) | dg (m) | M (kg) | AR |
|---|---|---|---|---|---|---|---|---|---|---|
| 1 | 37.7310 | 15.0078 | 2706 | 1809.0 | 0.090 | 0.065 | 0.060 | 0.071 | 0.119 | 0.667 |
| 2 | 37.7310 | 15.0079 | 2706 | 1806.8 | 0.085 | 0.075 | 0.060 | 0.073 | 0.104 | 0.706 |
| 3 | 37.7310 | 15.0081 | 2706 | 1811.6 | 0.105 | 0.075 | 0.065 | 0.080 | 0.170 | 0.619 |
| 4 | 37.7311 | 15.0083 | 2700 | 1812.3 | 0.110 | 0.080 | 0.070 | 0.085 | 0.174 | 0.636 |
| 5 | 37.7310 | 15.0083 | 2700 | 1824.4 | 0.085 | 0.075 | 0.050 | 0.068 | 0.710 | 0.588 |
| 6 | 37.7309 | 15.0083 | 2703 | 1832.9 | 0.075 | 0.050 | 0.045 | 0.055 | 0.058 | 0.600 |
| 7 | 37.7309 | 15.0083 | 2702 | 1836.2 | 0.080 | 0.055 | 0.045 | 0.058 | 0.124 | 0.563 |
| 8 | 37.7308 | 15.0083 | 2702 | 1843.7 | 0.090 | 0.065 | 0.050 | 0.066 | 0.110 | 0.556 |
| 9 | 37.7308 | 15.0082 | 2705 | 1843.1 | 0.073 | 0.070 | 0.055 | 0.066 | 0.102 | 0.753 |
| 10 | 37.7308 | 15.0082 | 2705 | 1843.1 | 0.065 | 0.055 | 0.055 | 0.058 | 0.057 | 0.846 |
| 11 | 37.7308 | 15.0082 | 2705 | 1843.1 | 0.070 | 0.055 | 0.040 | 0.054 | 0.047 | 0.571 |
| 12 | 37.7308 | 15.0082 | 2705 | 1843.1 | 0.060 | 0.043 | 0.045 | 0.049 | 0.050 | 0.750 |
| 13 | 37.7305 | 15.0080 | 2703 | 1864.2 | 0.166 | 0.110 | 0.100 | 0.122 | 0.420 | 0.602 |
| 14 | 37.7310 | 15.0081 | 2708 | 1815.8 | 0.055 | 0.045 | 0.030 | 0.042 | 0.025 | 0.545 |
| 15 | 37.7310 | 15.0081 | 2708 | 1815.8 | 0.065 | 0.045 | 0.042 | 0.050 | 0.310 | 0.646 |
| 16 | 37.7310 | 15.0081 | 2708 | 1815.8 | 0.055 | 0.040 | 0.035 | 0.043 | 0.021 | 0.636 |
| 17 | 37.7312 | 15.0081 | 2711 | 1792.6 | 0.135 | 0.110 | 0.100 | 0.114 | 0.437 | 0.741 |
| 18 | 37.7314 | 15.0080 | 2714 | 1769.9 | 0.155 | 0.115 | 0.095 | 0.119 | 0.497 | 0.613 |
| 19 | 37.7314 | 15.0080 | 2713 | 1773.0 | 0.140 | 0.120 | 0.110 | 0.123 | 0.516 | 0.786 |
| 20 | 37.7316 | 15.0072 | 2726 | 1727.1 | 0.090 | 0.075 | 0.065 | 0.076 | 0.108 | 0.722 |
| 21 | 37.7316 | 15.0072 | 2726 | 1727.1 | 0.045 | 0.040 | 0.030 | 0.038 | 0.030 | 0.667 |
| 22 | 37.7316 | 15.0072 | 2726 | 1727.1 | 0.085 | 0.070 | 0.070 | 0.075 | 0.102 | 0.24 |
| 23 | 37.7316 | 15.0072 | 2726 | 1727.1 | 0.080 | 0.050 | 0.045 | 0.056 | 0.088 | 0.563 |
| 24 | 37.7316 | 15.0072 | 2726 | 1727.1 | 0.055 | 0.045 | 0.035 | 0.044 | 0.043 | 0.636 |
| 25 | 37.7316 | 15.0072 | 2726 | 1727.1 | 0.100 | 0.075 | 0.065 | 0.079 | 0.142 | 0.650 |
| 26 | 37.7316 | 15.0072 | 2726 | 1727.1 | 0.043 | 0.035 | 0.025 | 0.034 | 0.021 | 0.581 |

**Table 1.** *Cont.*

| Ballistics | LAT | LON | ALT (m) | D (m) | a (m) | b (m) | c (m) | dg (m) | M (kg) | AR |
|---|---|---|---|---|---|---|---|---|---|---|
| 27 | 37.7315 | 15.0069 | 2722 | 1733.7 | 0.065 | 0.055 | 0.040 | 0.052 | 0.066 | 0.615 |
| 28 | 37.7315 | 15.0069 | 2722 | 1733.7 | 0.090 | 0.050 | 0.045 | 0.059 | 0.095 | 0.500 |
| 29 | 37.7315 | 15.0069 | 2722 | 1733.7 | 0.060 | 0.045 | 0.040 | 0.048 | 0.055 | 0.667 |
| 30 | 37.7315 | 15.0069 | 2722 | 1733.7 | 0.055 | 0.045 | 0.035 | 0.044 | 0.043 | 0.636 |
| 31 | 37.7314 | 15.0068 | 2721 | 1735.8 | 0.130 | 0.070 | 0.085 | 0.092 | 0.427 | 0.654 |
| 32 | 37.7315 | 15.0068 | 2723 | 1727.9 | 0.095 | 0.070 | 0.055 | 0.072 | 0.128 | 0.579 |
| 33 | 37.7315 | 15.0068 | 2723 | 1727.9 | 0.070 | 0.05 | 0.040 | 0.052 | 0.068 | 0.571 |
| 34 | 37.7315 | 15.0068 | 2723 | 1727.9 | 0.090 | 0.07 | 0.055 | 0.070 | 0.144 | 0.611 |
| 35 | 37.7315 | 15.0068 | 2723 | 1727.9 | 0.060 | 0.05 | 0,03 | 0.045 | 0.038 | 0.500 |
| 36 | 37.7316 | 15.0072 | 2638 | 1725.5 | 0.050 | 0.04 | 0.025 | 0.037 | 0.029 | 0.500 |
| 37 | 37.7316 | 15.0072 | 2638 | 1725.5 | 0.075 | 0055 | 0.040 | 0.548 | 0.077 | 0.533 |
| 38 | 37.7316 | 15.0072 | 2638 | 1725.5 | 0.070 | 0.055 | 0.045 | 0.056 | 0.077 | 0.643 |
| 39 | 37.7316 | 15.0072 | 2638 | 1725.5 | 0.055 | 0.050 | 0.035 | 0.046 | 0.032 | 0.636 |
| 40 | 37.7316 | 15.0072 | 2638 | 1725.5 | 0.050 | 0.040 | 0.035 | 0.041 | 0.031 | 0.700 |

Considering five blocks, we obtained an average density value approximating $\sim$1100 kg/m$^3$. In contrast, the aspect ratio (*AR*) values are between 0.5 and 0.9. Based on the scheme of Folk (1974), clasts can be classified as very elongated (*AR* < 0.6), elongated (0.6 < *AR* < 0.63), sub-elongated (0.63 < *AR* < 0.66), intermediate-form (0.66 < *AR* < 0.69) sub-equant (0.69 < *AR* < 0.72), equant (0.72 < *AR* < 0.75) and very equant (*AR* > 0.75). Most of the clasts analyzed have an aspect ratio between 0.5 and 0.6, thus corresponding to very elongate and elongate.

### 3.2. Analysis of the Lava Fountain Height Variation

The explosive activity at the SEC was observed by the INGV-OE monitoring camera network from the night of 21 February 2022 (Figure 5). Strombolian explosions evolved in lava fountains shortly after 11:10 UTC (Local time = UTC + 1), accompanied by significant tephra emission and rapid formation and growth of an eruptive column (Figure 5). During this phase, the IJR detected by the ENT camera reached heights between 300 to 1000 m above the crater rim, with a maximum outflow velocity of about 80 m/s. Between 11:50 and 12:35 UTC, the IJR heights reached about 3000 m above the volcano crater rim (Figure 6a). At this time, maximum outflow velocities, estimated using Equation (1) and the IJR height derived from the ENT camera, range from 150 to 250 m/s (Figure 6b). After 12:30 UTC, the activity gradually decreased, ceasing completely around 12:50 UTC, after a duration of 110 min. The average value of the IJR height was about 1185 m above the crater rim.

### 3.3. Eject! Results for the Paroxysmal Episode of 21 February 2022

Modeling of the trajectory of the clasts erupted during the paroxysmal activity of 21 February 2022, was carried out using the program Eject! The parameters fixed in all simulations are density, particle shape, wind speed and atmospheric properties. The shape of the clasts was approximated to a sphere because the measured aspect ratio shows that the particles are roughly regular. Information regarding weather data on February 21 was obtained from the forecast system provided to INGV-OE by ARPA of Emilia Romagna [16]. The forecast provides an average wind speed of 12 m/s near the summit craters (3000 m altitude) in the SE direction, a ground temperature of 15 °C, and a thermal gradient of about 6.5 °C/km. The parameters considered as variables are the velocity and angle of clast ejection, the elevation difference between the takeoff elevation and the deposition point, the diameter of the clasts, and the drag coefficient. Specifically, the initial outflow velocities for the lava fountaining of 21 February 2022 were considered in the range of 25–250 m/s, with the associated IJR heights ranging from $\sim$100 to $\sim$3500 m above the crater rim. Considering that clasts can leave at any point of the IJR, different takeoff point elevations were considered between the elevation of the crater rim of the SEC and the

maximum $H_{IJR}$. In fact, volcanological observations show that some ballistics could leave the volcanic jet at higher heights than the crater rim. The diameter of the clasts was also considered variable; in particular, two main diameter classes were selected at 0.08 and 0.1 m. Regarding the properties related to the air drag force, based on the shape of the blocks (e.g., spheres, cubes, projectile-shaped, etc.), there are different values of the drag coefficient experimentally obtained. Values of the drag coefficient considered in this work were constant and set to $C_d = 0.5$, 0.8 and 1.

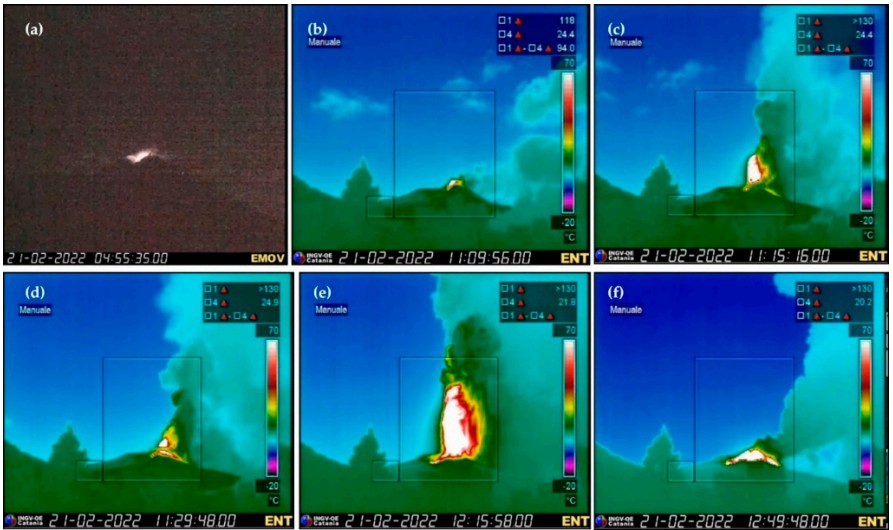

**Figure 5.** Photographs from the EMOV (Montagnola visible) and ENT (Nicolosi Thermal) cameras, showing the main phases of the eruptive activity that occurred on the morning of 21 February 2022; in panel (**a**), the beginning of Strombolian activity at the SEC can be seen as early as the previous night, and (**b**–**f**), the evolution of the lava fountain. The colored bar on the right side of each image shows the uncalibrated temperature between −20 and 70 °C. EMOV and ENT cameras belong to the surveillance system of INGV-OE (www.ct.ingv.it (accessed on 28 December 2022)).

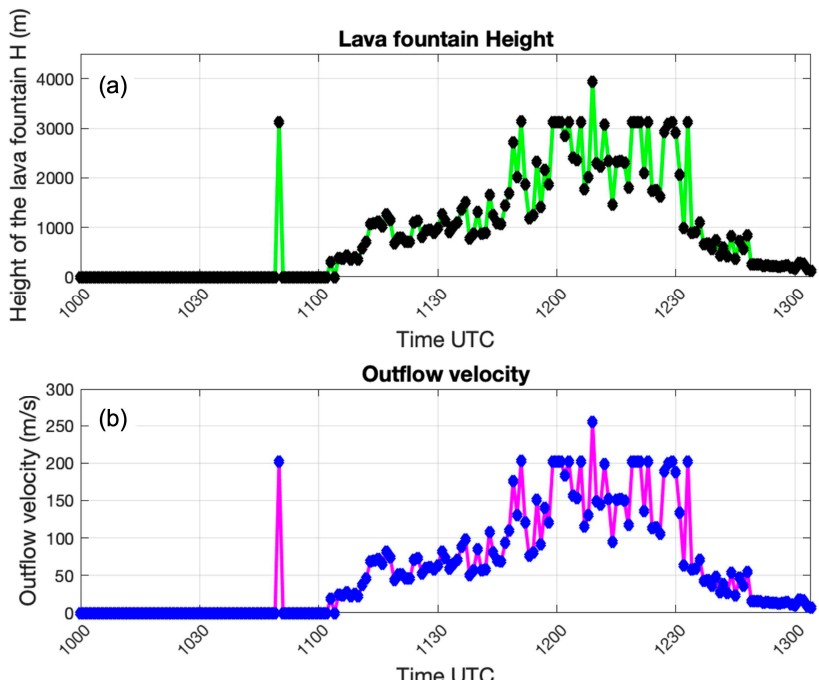

**Figure 6.** (**a**) Graph showing the height of the Incandescent Jet Region (IJR) of the 21 February 2022 lava fountain detected by the ENT camera of INGV-OE, as a function of time. (**b**) Graph showing the exit velocity estimated following [10] and using Equation (1), as a function of time.

The Eject! software allowed the reconstruction of the ballistic trajectories of the clasts ejected during the lava fountaining of 21 February 2022. Two rounds of tests were performed on the samples. During the first set (Table 2), simulations were carried out for each of the two selected diameter classes (0.08, and 0.1 m) and for drag coefficients of $C_d = 0.5$, $C_d = 0.8$, and $C_d = 1.0$, respectively (Figure 7). Progressively increasing outflow velocities and H$_{IJR}$ were also considered. The obtained results show that the outflow velocity and the distance of the landing point below the take-off point are in the range 150–220 m/s and 3000–3900 m respectively. However, it should be considered that from these height values the topographic elevation between the ground level and the SEC must be subtracted (∼3350–2750 m = ∼600 m), so the actual height of the IJR considered is between 2400 and 3300 m above the crater rim. Values obtained from the first set of tests are in very good agreement with the field data and were used to carry out a second round of simulations, varying the ejection angles. In this case, trajectories allowing the clasts to reach the real covered distance are obtained using angles between 20° and 50°.

**Table 2.** Input data used in Eject! Allowing the derivation of the output data compatible with distances actually reached by the clasts during the lava fountain of 21 February 2022.

| INPUT | | | | OUTPUT | | | | |
|---|---|---|---|---|---|---|---|---|
| Diameter Class (m) | Ejection Angle (Degree) | Initial Velocity (m/s) | Distance of Landing Point below the IJR (m) | Drag Coefficient | Distance (m) | Maximum Height (m) | Final Velocity (m/s) | Travel Time (s) |
| 0.08 | 45 | 155 | 3300 | 0.5 | 1762 | 318 | 70 | 60 |
| 0.08 | 45 | 180 | 3700 | 0.8 | 1768 | 319 | 57 | 75 |
| 0.08 | 45 | 200 | 3800 | 1.0 | 1747 | 297 | 51 | 83 |
| INPUT | | | | OUTPUT | | | | |
| Diameter Class (m) | Ejection Angle (Degree) | Initial Velocity (m/s) | Distance of Landing Point below the IJR (m) | Drag Coefficient | Distance (m) | Maximum Height (m) | Final Velocity (m/s) | Travel Time (s) |
| 0.1 | 45 | 120 | 3000 | 0.5 | 1773 | 319 | 78 | 52 |
| 0.1 | 45 | 110 | 3400 | 0.8 | 1730 | 317 | 63 | 65 |
| 0.1 | 45 | 90 | 3700 | 1.0 | 1768 | 319 | 57 | 75 |

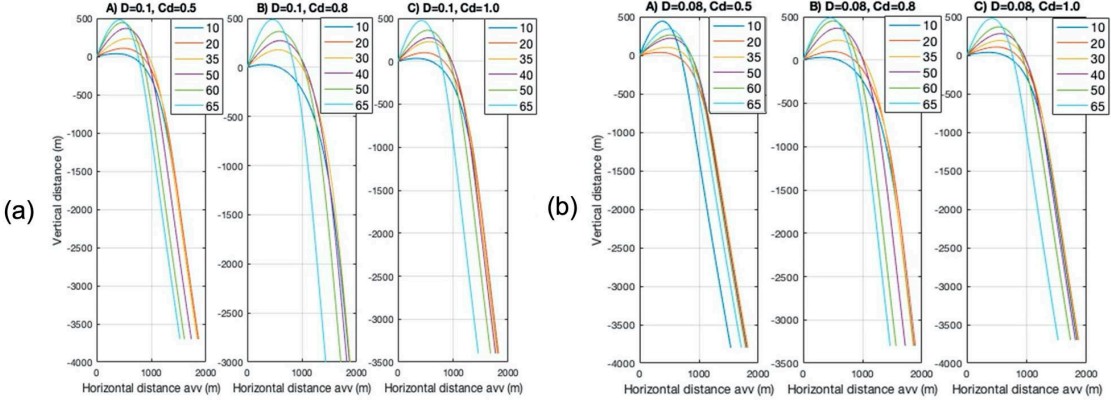

**Figure 7.** Trajectories of ballistics obtained with the Eject! software respectively for different drag coefficients and sizes. Different colors represent different angles. Diagrams reported in this figure define the ballistic trajectories as a function of the height difference between the ground level and the IJR (i.e., the vertical distance above the vent) and the horizontal distance from the vent for a dimeter of (**a**) 0.1 m and (**b**) 0.08 m.

*3.4. Eject! Results for Other Paroxysmal Episodes*

Although the general behavior characterizing the paroxysmal eruptions at Etna is largely comparable, it is worth noting that the intensity of the eruptive activity and the

meteorological conditions (e.g., wind speed) could vary considerably. Both factors mainly control the ballistic trajectories and the distance reached.

For this reason, we have selected four other episodes throughout the 2020–2022 paroxysmal sequence, which were characterized by different eruptive and meteorological conditions with respect to the 21 February 2022 episode.

Moreover, during those episodes, the fallout of large clasts in touristic areas was also reported. The paroxysmal episodes considered are those of 18, 23 and 28 February and 23 October 2021 (Figure 8). Figure 9 shows the tephra fallout deposit forecasts daily run at INGV-OE.

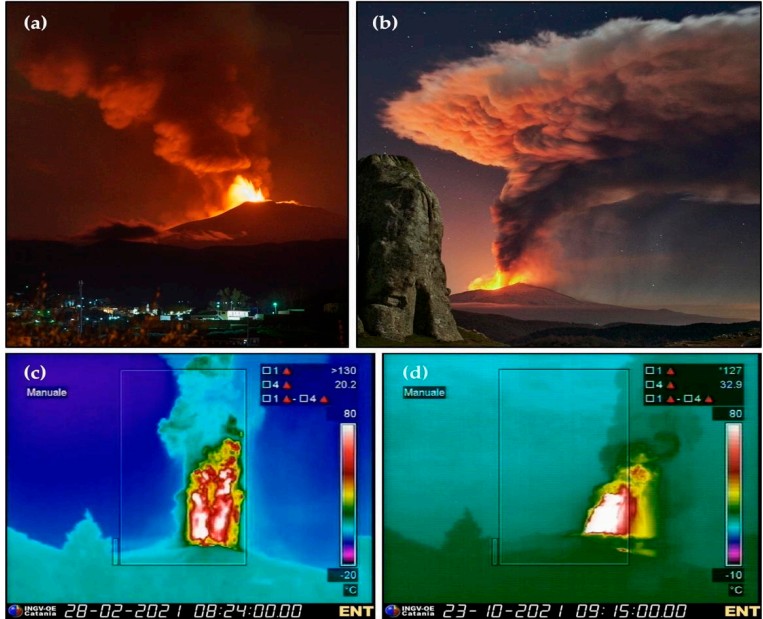

**Figure 8.** The four paroxysmal episodes examined to carry out the second set of simulations with Eject! The paroxysmal episode of (**a**) 18 February 2021 (photo by G. Costa); (**b**) 23 February 2021 (photo by G. Costa); (**c**) Image of the lava fountain on 28 February 2021 from the TIR camera; (**d**) Image of the lava fountain of 23 October 2021 from the ENT camera. TIR and ENT cameras belong to the surveillance system of INGV-OE (www.ct.ingv.it (accessed on 28 December 2022)).

For these lava fountains, a series of tests using the Eject! software has been performed in order to evaluate how the trajectory and distance reached by clasts greater than 5 cm in size change as eruptive and meteorological conditions vary. In fact, during those events, some tourist guides observed the fallout of centimeter-sized clasts at distances greater than 5 km from the SEC.

Localities affected by the fallout of large clasts considered in this study are (Figure 1): Piano del Vescovo on February 18 (located at a distance of about 7 km from the SEC), Monte Rittmann on February 23 (located at a distance of 2.7 km from the SEC), Monte Rinatu on February 28 (located at a distance of about 6 km from the SEC) and Rifugio Citelli on October 23 (at a distance of about 5 km from the summit). Since direct analyses were not carried out on samples produced by these paroxysmal episodes, parameters fixed as constant in these cases include the drag coefficient, which was set at a value of 0.1 with an artillery shell type clast shape associated because we obtained the maximum distance reached by ballistics. Regarding the properties of the clasts, the diameter considered was 10 cm. In this case, atmospheric properties were obtained from weather data provided by ARPA [16].

The other variable parameters for the modeling are the velocity and angle of clast outflow and the distance between the IJR and the point of fall. Finally, the $H_{IJR}$ and the relative velocities of outflow are known from the analyses performed with the ENT thermal camera (Figure 10).

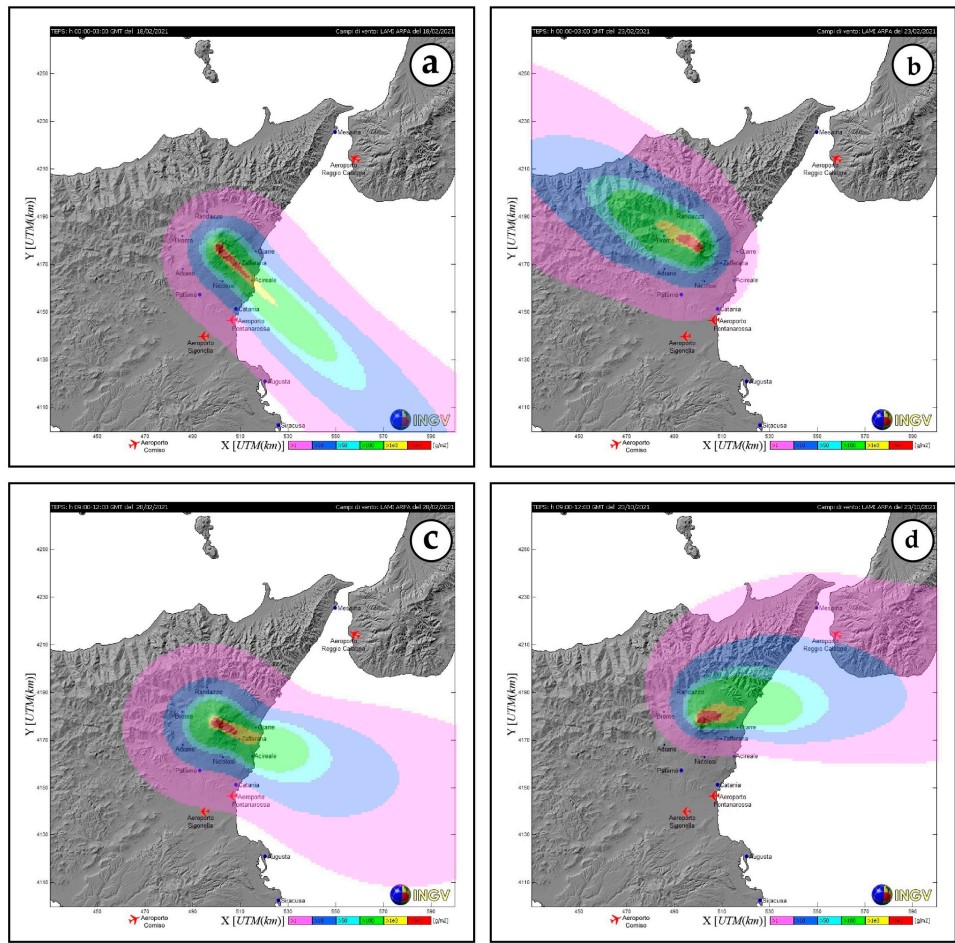

**Figure 9.** Tephra fallout computed by INGV-OE during the paroxysmal episode of: (**a**) 18 February 2021, (**b**) 23 February 2021, (**c**) 28 February 2021, and (**d**) 23 October 2021.

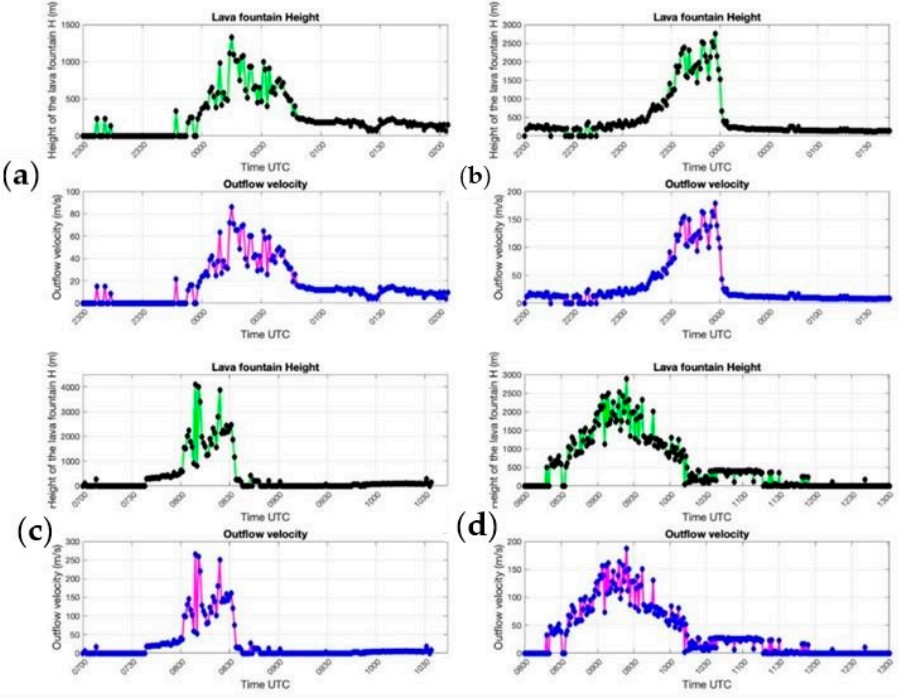

**Figure 10.** Graph showing the height of the Incandescent Jet Region (IJR) and the exit velocity for (**a**) 18 February 2021, (**b**) 23 February 2021, (**c**) 28 February 2021, and (**d**) 23 October 2021.

For each lava fountaining episode, we performed different sets of simulations modifying the model input parameters and comparing distances reached by ballistics with those found in the field (Table 3).

**Table 3.** Input data used in Eject! allowing the derivation of the output data most compatible with the distances actually reached by the clasts during the lava fountaining episodes of 18, 23 and 28 February and 23 October 2021.

| | INPUT | | | | | OUTPUT | | | | |
|---|---|---|---|---|---|---|---|---|---|---|
| Lava Fountain | Diameter Class (m) | Ejection Angle (Degree) | Initial Velocity (m/s) | Distance of Landing Point below the IJR (m) | Drag Coefficient | Density (kg/m³) | Distance (m) | Maximum Height (m) | Final Velocity (m/s) | Travel Time (s) |
| Feb 18 | 0.1 | 45 | 200 | 4720 | 0.1 | 1300 | 7066 | 923 | 266 | 50 |
| Feb 23 | 0.1 | 70 | 180 | 2950 | 0.1 | 1500 | 2721 | 1272 | 229 | 47 |
| Feb 28 | 0.1 | 70 | 280 | 5700 | 0.1 | 1500 | 5936 | 2935 | 295 | 70 |
| Oct 23 | 0.1 | 50 | 230 | 4250 | 0.1 | 1500 | 5459 | 1173 | 195 | 54 |

## 4. Discussion

Assessment of the eruption source parameters during a volcanic eruption is important for modeling accurately the ballistic trajectories of the emitted clasts and evaluating in turn the potential effects caused by such kinds of events. During the 2020–2022 paroxysmal sequence, some lava fountains reached considerable, although variable, heights and intensities, occurring under largely different wind speeds. Eruptive parameters, together with data obtained on the field, were included in the modeling to estimate the maximum distance reached by the ballistics. Clast diameter and density are important factors in determining the maximum distance reached by ballistics. However, it is noteworthy that field sampling carried out soon after the eruptive event is not always possible, especially when the occurrence of events is very fast.

In this paper we show how, coupling a simple ballistic software with some eruptive parameters retrieved by remote sensing systems, can be used to estimate the area affected by the fallout of ballistics. However, our methodology has some limitations. Results of the simulations carried out with Eject! refer to individual clasts, but probably during lava fountain activity the interactions between blocks continuously could influence the trajectory, size and, as a consequence, the fallout of clasts [27]. In addition, simulations performed for this study assume that the takeoff elevation during lava fountaining activity can occur within the whole jet using the velocity estimated at the crater rim, while it is well known that the clast velocity decreases with the height [28]. However, analyzing the best agreement among simulations and field data of the 21 February 2022 eruption, we found a maximum ejection height between 2500 and 3300 m that is within the IJR (Figure 5). We expected that the initial velocity should be lower than the value estimated using images of the thermal cameras. However, the maximum speed of IJR allowed the identification of a wider area subjected to the fallout for those ballistics, a feature potentially having implications for the delimitation of restricted areas in order to mitigate risks for the population. Our results show some discrepancies in matching the data actually observed. In fact, the results acquired using the IJR outflow velocity and HIJR calculated through the ENT camera for the episode of 18 February 2021 show an underestimation of the distances really reached by the clasts. Clasts considered in our study are in the range of smaller ballistics having a size of about 0.1 m. Consequently, these clasts can be subjected to strong air drag, which could have been greatly reduced if they were flying behind other larger clasts. Moreover, this could be related to the incorporation of a number of clasts as large as >5–10 cm within the eruptive column, which were transported up to the higher portions of the eruptive column, where they are most affected by the winds and can precipitate at larger distances from the point of emission [6,14]. Those particles cannot

be considered as ballistics requiring a different modeling approach. In particular, the risk associated with the fallout of large clasts from the convective portion of an eruptive column is often overlooked, although field evidence, especially in the case of Mt. Etna, clearly shows that this is a common feature for both low and high-intensity eruptions [14]. Finally, a comparative study between other models which simulates ballistic trajectories [29] could be useful in the future. The use of different models, as widely tested to compute the tephra fallout [30] could help to better quantify the uncertainty of the hazard associated with those events. Furthermore, as in some cases, some particles having a size > 5 cm could reach the convective regions, thus increasing the distance reached from the summit craters [14], and for this reason, the use of more complex models also coupled with the lava fountain [31] should be valuable.

## 5. Conclusions

The hazard associated with the fallout of ballistics is becoming an issue progressively more frequent throughout the recent eruptive record of Etna. Particularly during the latest paroxysmal sequence, composed of more than sixty episodes of lava fountaining that took place between mid-December 2020 and February 2022, significant fallout of large clasts (>5 cm) occurred several kilometers far away the SEC, affecting areas often visited by large numbers of tourists. In this study, we have presented a set of simulations of the ballistic trajectories drawn by the erupted products. The model has been used through the comparison with data taken during a series of field surveys and integrating weather data and the relative $H_{IJR}$ during the paroxysmal phase of the lava fountain. These data provide the eruptive conditions necessary for simulating the ballistic trajectory for the clasts found at the real sampling sites. Future studies could be conducted by integrating results from more sophisticated models. Finally, knowing the height of the fountain and the intensity of winds, a real-time model could be implemented to assess the impact produced by a lava fountain episode. This is the first step in creating a real-time and free available system capable of assessing the possible impact during paroxysms in order to mitigate the risk associated with the fallout of large ballistics, especially close to areas densely affected by tourists and hikers.

**Author Contributions:** Conceptualization, S.S. and M.V.; methodology, S.S. and M.V.; software, G.C. and L.M.; validation, G.C., L.M., M.P., S.S. and M.V.; formal analysis, G.C., L.M., M.P., S.S. and M.V.; investigation, G.C., L.M., M.P, S.S. and M.V.; resources, S.S. and M.V.; data curation, G.C., S.S. and M.V.; writing—original draft preparation, G.C.; writing—review and editing, G.C., L.M., M.P., S.S. and M.V.; visualization, G.C. and L.M.; supervision, S.S. and M.V.; project administration, S.S. and M.V.; funding acquisition, S.S. and M.V. All authors have read and agreed to the published version of the manuscript.

**Funding:** This project has received funding from the Project MIUR 2020–2029 "Pianeta Dinamico-PANACEA" (CUP D53J19000170001).

**Data Availability Statement:** The camera data used in this study are property of INGV-OE (Istituto Nazionale di Geofisica e Vulcanologia—Osservatorio Etneo, Sezione di Catania). They can be made available, upon reasonable request, asking them to the corresponding author.

**Acknowledgments:** The authors would like to thank the technicians from INGV-OE for monitoring the camera network maintenance, especially E. Pecora, E. Biale, P. Principato and F. Ciancitto. This work was performed in the framework of the INGV Project "Pianeta Dinamico" (D53J19000170001), funded by MUR ("Ministero dell'Università e della Ricerca, Fondo finalizzato al rilancio degli investimenti delle amministrazioni centrali dello Stato e allo sviluppo del Paese, legge 145/2018").

**Conflicts of Interest:** The authors declare no conflict of interest.

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
