# Peer review of "Modeling the Trajectories of Ballistics in the Summit Area of Mt. Etna (Italy) during the 2020–2022 Sequence of Lava Fountains"

_geosciences, doi:10.3390/geosciences13050145_

Round 1

Reviewer 1 Report

This paper analyzes some important characteristics of a paroxysmal eruption at Mount Etna that occurred on 21 February 2022.  In particular, it uses the height of lava fountains to estimate exit velocity, and measures a few dozen ballistic blocks in the field, including for example size, shape, weight, and landing location.   These properties were then fed into the ballistic program Eject to estimate the conditions of ejection; exit velocity, trajectory angle, and drag properties, that could have deposited the blocks at the observed locations.  Similar calculations were made (I think) for hypothetical blocks during paroxysms on 23 February, 28 February, and 23 October. 

Ballistic block ejection is an important hazard and Etna, and studies of this type will help identify the conditions under which block ejection could be a hazard.  For this reason, I think that a paper on this topic would be of significant interest.  However, I have several major concerns with this manuscript.

1.       In several places, the English is garbled to the point where the meaning is unclear.  This problem is obvious for example in Section 2.2, describing the assumptions used in equations; and Section 3.3, describing the inputs to the Eject model and their basis. 

2.       In several other places, things just aren’t adequately explained.  In Section 3.4, it wasn’t clear to me what was actually done at the extra locations identified.  Were these hypothetical landing sites for modeled blocks?  In figures 7 and 8, the meaning of different curves in each sub-figure are not explained.

3.       The ballistics model Eject is not the appropriate tool for this problem in my opinion.  The Eject model assumes that the blocks are traveling through still air (or air with a constant head- or tailwind), which is appropriate for big blocks ejected in impulsive eruptions.  The diameters of blocks modeled in this study (Table 1) are only a few centimeters, and are ejected in a steady jet. Rising gas in the jet will lift the blocks, especially if they are small.    Steady exit velocities of these lava fountains for example were 150-250 m/s (Fig. 5).  The settling velocity of a 6-cm diameter block having an average density of 1,000 kg/m3 in air at 300 K would be about 40 m/s. 

4.       Some ad-hoc adjustments have been made that mitigate the limitations of the model, but in my opinion are not appropriate.  For example, the takeoff elevation is taken to be the top of the lava fountain.  But the initial velocity is taken to be that inferred at the base of the fountain base on IJR height.  at the top of the fountain, the ejected material would likely have a vertical velocity component of zero.

5.       The results of the ballistics calculations don’t add a lot of insight into the conditions that caused hazardous ballistic ejection.  The Discussion section for example concludes that four factors (drag coefficient, wind, fountain height, ejection angle) influence final distance, and that distances calculated using IJR exit velocity and fountain height underestimate the actual observed distances.  The four factors could have been inferred without external data, and the final point in my opinion is not valid because it’s unreasonable to use the IJR exit velocity and fountain height as input values.

In summary, I don’t think that this paper, using Eject calculations, can easily be made publishable using the current particle sizes and inputs.  Attempting to show through calculations that the approach using Eject provides a realistic particle trajectory would require more sophisticated modeling and essentially make the Eject calculations unnecessary.  It may be possible to use the ejection velocities, as done by others for other eruptions at Etna.  However, such a paper would be substantially different and couldn’t be considered a revision of this one.

I’m sorry, but I don’t see any clear way to make this paper publishable.

Detailed and specific comments 

Please include line numbers!

Section 2.1: 

There are some important details missing in this explanation of field data collection.  For example, how was the weight determined?  Did you take a scale into the field and weigh them there, or did you collect the samples, and bring them to the lab to weigh them?  What was the accuracy of the scale?  1g?  0.1g?  If you weighed the samples in the field, did you shield the scale from wind?  how is density calculated?

Section 2.2, first paragraph:  This description of the three parts of an eruptive column is not really accurate.  I think you are referring to the (1) jet thrust region, where jet density is greater than air; (2) the convective thrust region, where density is less than air, and (3) the overshooting top, where density is again greater than air.

Paragraph 3:  what does mK stand for?  Millikelvins?

Paragraph 4:  This description of assumptions used in the Torricelli equation are unclear.  What does “projectiles under uniformly accelerating motion” mean?  Gravitational acceleration is constant.  Are you saying that the gravitational force is the only force acting on these particles?  No air drag?  You say in the last sentence that “changes in . . . air drag are neglected”.  Do you mean that air drag is neglected?  It would help to present the Torricelli equation here.  Neither of the papers you cite here use the term “Torricelli equation”, so it’s unclear which equation  you’re using.

Equation 1:  there appear to be two equations on this line.  The first should be crossed out, so that eq. 1 reads H_IJR=v_ex^2/(2g).

Last sentence of section 2.2:  What method was used to convert the height in pixels to meters?

Section 3.1

First paragraph.  You need to cite Table 1 for the data that are described in this paragraph.

Section 3.2:

change “UTC (Local time = +1)” to “UTC (Local time=UTC+1)”

Section 3.3:

The input parameters used in the modeling also appear to be unclearly described, and in some cases inappropriate.  For example, it took me a few readings of the first paragraph of this section to realize that you are using the IJR height for the takeoff elevation of the block.  Later, in the discussion section, you say that you sometimes used the IJR velocity inferred from the fountain height as the starting velocity.  The takeoff location is referred to as the “zone of fragmentation”, which is poorly defined.  The IJR velocity would be inappropriate to use for a block that starts at the IJR height.  And at the IJR height, the vertical component of velocity should be essentially zero.

Section 3.4:

Paragraph 2:  What exactly are these locations that you mention at the beginning of this paragraph?  Hypothetical landing sites for modeled blocks?  After reading this paragraph three or four times, it appears that that’s what you mean, but I’m not sure.  On Figure 1, these locations are indicated with yellow stars, same as the SEC.  If they’re not vents, depicting them using a different symbol would make it clear.  Also, a legend should be added to Figure 1.

Discussion (Section 4.)

The last paragraph contains important information on the hazards of ballistics at Etna.  It would be useful in a future paper to put this information in the Introduction.

Figures:

Figure 1b:  this isn’t a very clear map of the field area.  For example, at this scale, can the vent on this map be considered a point source?  Is it a fissure or a central vent?  In addition, if you’re using ballistics calculations, I would expect to see a map showing both the block locations and the vent location.

Figure 2:  I assume that the graduations on your folding ruler are in centimeters, but it would be important to indicate that.  On Figures 2c and 2d, the centimeter markings aren’t visible, making the ruler useless as a scale.  If you indicate the length of each folding segment, that distance could be used as a scale.

Tables:

Table 1:  Perhaps I missed it, but I don’t see a vent location given, in latitude  and longitude.  I presume you would need this in order to calculate the distance of travel of the clasts.  Also, did you calculate clast volume?  That should be included if it was used to obtain clast density.  Can you justify six significant figures of accuracy in some of the clast density measurements?

Figure 4 caption:  change “in panels (a)” to “in panel (a)”.  And change “and (b) (c), (d) (e) (f)” to “and in panels (b), (c), (d), (e), and (f)”

Figure 5: change “outflow velocity” to “exit velocity”

Author Response

Dear Editor

We are submitting the revised version of the paper entitled “Hazard assessment by ballistics in the summit area of Mt. Etna during the 2021-22 lava fountains” by Costa et al. We thank the reviewers for the useful comments and suggestions and believe that now this manuscript can be considered for publication by Geoscience.  We would like to thank the reviewers for their comments that helped us to greatly improve the paper. Please, find hereafter our replies to all their comments. Thank you for your consideration of this manuscript.

Sincerely,

Simona Scollo on behalf of all the authors

Reviewer 1:

  1. In several places, the English is garbled to the point where the meaning is unclear.  This problem is obvious for example in Section 2.2, describing the assumptions used in equations; and Section 3.3, describing the inputs to the Eject model and their basis. 

The English language was improved and many parts of the text were re-written.

  1. In several other places, things just aren’t adequately explained.  In Section 3.4, it wasn’t clear to me what was actually done at the extra locations identified.  Were these hypothetical landing sites for modeled blocks?  In figures 7 and 8, the meaning of different curves in each sub-figure are not explained.

We explained how the extra locations were identified. Figure 7 and 8 were re-made and better explained.

  1. The ballistics model Eject is not the appropriate tool for this problem in my opinion.  The Eject model assumes that the blocks are traveling through still air (or air with a constant head- or tailwind), which is appropriate for big blocks ejected in impulsive eruptions.  The diameters of blocks modeled in this study (Table 1) are only a few centimeters, and are ejected in a steady jet. Rising gas in the jet will lift the blocks, especially if they are small.    Steady exit velocities of these lava fountains for example were 150-250 m/s (Fig. 5).  The settling velocity of a 6-cm diameter block having an average density of 1,000 kg/m3 in air at 300 K would be about 40 m/s. 

We are considering the fallout of the particles outside the volcanic jet when they are affected by the only ballistics contribution. The possible interaction with fluid jets is not here considered.

  1. Some ad-hoc adjustments have been made that mitigate the limitations of the model, but in my opinion are not appropriate.  For example, the takeoff elevation is taken to be the top of the lava fountain.  But the initial velocity is taken to be that inferred at the base of the fountain base on IJR height.  at the top of the fountain, the ejected material would likely have a vertical velocity component of zero.

We are not considering the take-off elevation to be the top of the lava fountain but we are changing the take-off elevation between the crater rim and the top. We know that the vertical component of jet velocity decreases in function of height, as reported in literature (Taddeucci et al., 2015). However, considering the velocity obtained from the thermal camera we are in the worst condition. Consequently, the fallout of ballistics beyond the area estimated by the model is not expected.

  1. The results of the ballistics calculations don’t add a lot of insight into the conditions that caused hazardous ballistic ejection.  The Discussion section for example concludes that four factors (drag coefficient, wind, fountain height, ejection angle) influence final distance, and that distances calculated using IJR exit velocity and fountain height underestimate the actual observed distances.  The four factors could have been inferred without external data, and the final point in my opinion is not valid because it’s unreasonable to use the IJR exit velocity and fountain height as input values.

In this paper we tested that a ballistic model can be applied also for lava fountain events and extended this study to other events that occurred at Etna. Whereas the model and what input parameters mainly control the fallout of ballistics has been widely investigated, similar models have never applied at Etna. Following the suggestion of the reviewer, we have improved the discussion section. 

Detailed and specific comments. Please include line numbers!

We are sorry, the line numbers were included.

  • Section 2.1: There are some important details missing in this explanation of field data collection.  For example, how was the weight determined?  Did you take a scale into the field and weigh them there, or did you collect the samples, and bring them to the lab to weigh them?  What was the accuracy of the scale?  1g?  0.1g?  If you weighed the samples in the field, did you shield the scale from wind?  how is density calculated?

Done, we added how the weight and density were estimated.

  • Section 2.2, first paragraph:  This description of the three parts of an eruptive column is not really accurate.  I think you are referring to the (1) jet thrust region, where jet density is greater than air; (2) the convective thrust region, where density is less than air, and (3) the overshooting top, where density is again greater than air.

The reviewer is right; the sentences were deleted.

  • Paragraph 3:  what does mK stand for?  Millikelvins?

Yes, the unit was modified in International standard.

  • Paragraph 4:  This description of assumptions used in the Torricelli equation are unclear.  What does “projectiles under uniformly accelerating motion” mean?  Gravitational acceleration is constant.  Are you saying that the gravitational force is the only force acting on these particles?  No air drag?  You say in the last sentence that “changes in . . . air drag are neglected”.  Do you mean that air drag is neglected?  It would help to present the Torricelli equation here.  Neither of the papers you cite here use the term “Torricelli equation”, so it’s unclear which equation you’re using.

The reviewer is right. We are considering the Torricelli for a no-viscous flow. This methodology, although it is an approximation, has been successfully applied at Etna by other authors (15, 16).

  • Equation 1:  there appear to be two equations on this line.  The first should be crossed out, so that eq. 1 reads H_IJR=v_ex^2/(2g).

We are sorry, it was a typo and the equation is now correct.

  • Last sentence of section 2.2:  What method was used to convert the height in pixels to meters?

We added some additional information and a reference [15].

  • Section 3.1: First paragraph.  You need to cite Table 1 for the data that are described in this paragraph.

Done.

  • Section 3.2: change “UTC (Local time = +1)” to “UTC (Local time=UTC+1)”
    Done.
  • Section 3.3: The input parameters used in the modeling also appear to be unclearly described, and in some cases inappropriate.  For example, it took me a few readings of the first paragraph of this section to realize that you are using the IJR height for the takeoff elevation of the block.  Later, in the discussion section, you say that you sometimes used the IJR velocity inferred from the fountain height as the starting velocity.  The takeoff location is referred to as the “zone of fragmentation”, which is poorly defined.  The IJR velocity would be inappropriate to use for a block that starts at the IJR height.  And at the IJR height, the vertical component of velocity should be essentially zero.

The reviewer is right. We are not considering the fragmentation processes but we are hypnotizing that takeoff elevation could occur with the entire lava fountain altitude. The ‘zone of fragmentation’ was replaced with the takeoff elevation.

  • Section 3.4: Paragraph 2:  What exactly are these locations that you mention at the beginning of this paragraph?  Hypothetical landing sites for modeled blocks?  After reading this paragraph three or four times, it appears that that’s what you mean, but I’m not sure.  On Figure 1, these locations are indicated with yellow stars, same as the SEC.  If they’re not vents, depicting them using a different symbol would make it clear.  Also, a legend should be added to Figure 1.

We added why these locations were considered. Moreover, Figure 1 was also modified.

  • Discussion (Section 4.): The last paragraph contains important information on the hazards of ballistics at Etna.  It would be useful in a future paper to put this information in the Introduction

Done. The last sentence was moved to the introduction.

Figures:

  • Figure 1b:  this isn’t a very clear map of the field area.  For example, at this scale, can the vent on this map be considered a point source?  Is it a fissure or a central vent?  In addition, if you’re using ballistics calculations, I would expect to see a map showing both the block locations and the vent location.

Done, following the suggestion of the reviewer, Figure 1 was modified. 

  • Figure 2:  I assume that the graduations on your folding ruler are in centimeters, but it would be important to indicate that.  On Figures 2c and 2d, the centimeter markings aren’t visible, making the ruler useless as a scale.  If you indicate the length of each folding segment, that distance could be used as a scale.

Done, figure 2 (now Figure 3) was modified.

  • Figure 4 caption:  change “in panels (a)” to “in panel (a)”.  And change “and (b) (c), (d) (e) (f)” to “and in panels (b), (c), (d), (e), and (f)”
  • Figure 5: change “outflow velocity” to “exit velocity”.

Done

Tables:

  • Table 1:  Perhaps I missed it, but I don’t see a vent location given, in latitude and longitude.  I presume you would need this in order to calculate the distance of travel of the clasts.  Also, did you calculate clast volume?  That should be included if it was used to obtain clast density.  Can you justify six significant figures of accuracy in some of the clast density measurements?

The vent location was added in Figure 1. Moreover, we added how the clast density was estimated. We delete the column of the volume as it was estimated using the approximation of an ellipsoid based on three main axes and it was not used in text.

Reviewer 2 Report

The authors explore the trajectory and associated parameters of ballisitics from Etna lava fountains, which indeed raise a main question for hazard assessment in this volcanic area. They use and compare field data and numerical simulation using an open-source software. The manuscript is well written and the authors discuss the implications of the results presented but also their limits. In conclusion, this paper seems to be a great work for the scientifc/civil community that is more and more interested in volcanic hazards assessment. Therefore, I recommend publication of the manuscript after minor revisions that are noted in the attached pdf file. My only comments are typos, maybe a lack of a few references for some scientific statements, and a lack of explanation for the clast density determination.

Author Response

Dear Editor

We are submitting the revised version of the paper entitled “Hazard assessment by ballistics in the summit area of Mt. Etna during the 2021-22 lava fountains” by Costa et al. We thank the reviewers for the useful comments and suggestions and believe that now this manuscript can be considered for publication by Geoscience.  We would like to thank the reviewers for their comments that helped us to greatly improve the paper. Please, find hereafter our replies to all their comments. Thank you for your consideration of this manuscript.

Sincerely,

Simona Scollo on behalf of all the authors

Reviewer 2:

  1. Section 1:
    1. “blocks and scoria even of centimeter size “ ? Blocks define a type of pyroclast from its size and shape, while scoria define a type of pyroclast from its texture. I would suggest to rephrase this part

Done.

  1. Show “Barbagallo Craters” in Fig. 1b.

Done.

  1. Section 2.1: “Also the relative density has been calculated.” ? I would suggest to precise/explain the method used for density determination. If it has been determined in laboratory as suggested in the abstract, I would also suggest to modify the title 2.1 in order to add laboratory analysis.

Done.

  1. Section 3.3: “However, ejection angles can vary considerably, so values ranging from 20°up to 80° were considered.” ? How do you consider this range? Do you have some visual insights, previous works, ect..?
    The sentence was deleted. In fact, we have analyzed all the angles.

Reviewer 3 Report

This article presents a ballistic study about fragments ejected during a lava fountain event at Mt. Etna volcano on 21 February 2022, including field data of several samples and modeling of their trajectories using a simple ballistic calculator, combined with some observations from a thermal camera. Given the relevance of the ballistic hazard at MT. Etna and the utility of the field data and thermal observations, I think that this study deserves to be published in Geosciences.

However, the manuscript must be significantly improved before publication and several major issues should be addressed:

1) Title. This is not really a hazard assessment study, which would result for instance, in a hazard map or probabilistic results for ballistic impacts. Instead, this is a ballistic study of the fragments ejected during a specific lava fountain event.

2) There are some important inconsistencies in the fragment densities between the values measured in the field and those used in the simulation in the different scenarios.

3)      Instead of presenting the captures of the eject interphase, the authors should make their own plots showing the different trajectories that they modeled with the different conditions, using an appropriate scaling to better compare the different trajectories.

4)      At the end of section 3.3 (p.8), it says that a second round of simulation varying the ejection angle was performed. I could not find the results of these simulations in the manuscript. They could be presented for instance in a plot showing the horizontal range vs ejection angle for different conditions.

5)      It is not clear whether the authors have evidence of ballistic fragments reaching the localities mentioned in section 3.4 on the specified dates, or if these are just hypothetic scenarios.

6)      The ballistics modeled in section 3.4 reached greater distances than those observed in the field from the February 21, 2022 event. Are these differences real (i.e. based on data), or result only from the differences in the drag coefficient value considered?

7)      The limitation of the Eject model should be discussed in more detail. See for instance ref. 27.

8)      Overall, the manuscript should be carefully revised and the English language and style improved.

In the attached pdf I included more detailed comments and suggestions that must be revised before publication.

Author Response

Dear Editor

We are submitting the revised version of the paper entitled “Hazard assessment by ballistics in the summit area of Mt. Etna during the 2021-22 lava fountains” by Costa et al. We thank the reviewers for the useful comments and suggestions and believe that now this manuscript can be considered for publication by Geoscience.  We would like to thank the reviewers for their comments that helped us to greatly improve the paper. Please, find hereafter our replies to all their comments. Thank you for your consideration of this manuscript.

Sincerely,

Simona Scollo on behalf of all the authors

Reviewer 3:

1) Title. This is not really a hazard assessment study, which would result for instance, in a hazard map or probabilistic results for ballistic impacts. Instead, this is a ballistic study of the fragments ejected during a specific lava fountain event.

Thanks for this comment. The title was modified following the suggestion of the reviewer. 

2) There are some important inconsistencies in the fragment densities between the values measured in the field and those used in the simulation in the different scenarios.

The reviewer is right. We measured only five clasts in the laboratory and this density was considered in the simulation. We deleted the density estimated in the table, estimated considering a volume equal to an ellipse with a,b, c measured in the field and laboratory.

3)  Instead of presenting the captures of the eject interphase, the authors should make their own plots showing the different trajectories that they modeled with the different conditions, using an appropriate scaling to better compare the different trajectories.

Done.

4)      At the end of section 3.3 (p.8), it says that a second round of simulation varying the ejection angle was performed. I could not find the results of these simulations in the manuscript. They could be presented for instance in a plot showing the horizontal range vs ejection angle for different conditions.

Done.

5)    It is not clear whether the authors have evidence of ballistic fragments reaching the localities mentioned in section 3.4 on the specified dates, or if these are just hypothetic scenarios.

Localities were chosen because the fallout of large clasts was observed during those events by natural and volcanological guides.

6)      The ballistics modeled in section 3.4 reached greater distances than those observed in the field from the February 21, 2022 event. Are these differences real (i.e. based on data), or result only from the differences in the drag coefficient value considered?

We observed greater clasts on 18 February in Piano del Vescovo; information of the fallout of ballistics on 23 and 28 February at Serracozzo was given by a volcanological and naturalistic guides respectively, whereas, for the event of 23 October, some information was obtained by internet. 

7)   The limitation of the Eject model should be discussed in more detail. See for instance ref. 27.

We improved the part concerning the limitation of our methodology (not for the modeling already discussed in previous papers).

8)  Overall, the manuscript should be carefully revised and the English language and style improved.

Done.

Detailed and specific comments.

  • Section 2.1: how was calculated density?

The estimation of the density is now added in the paper.

  • Section 2.2: What about the ash? it should also travel through this zone

We added two new figures including volcanic ash dispersal forecasts run at INGV-OE on a daily basis.

  • Section 2.3: “Different types of clast shapes, denoted as sphere, cube and artillery shell can be also considered”. ? These options are not only for calculating the average diameter, but to calculate a variable drag coefficient throughout the trajectory.

 We have added this explanation.

  • Section 3.1: The average of the densities presented in Table 1 seems to be significantly lower than 1000 kg/m3. I suggest calculating the mean value and the standard deviation of the measured data.

The reviewer is right, the density considering the mean value of five clasts collected was contemplated in the simulations.

  • Section 3.2: did the jet height increased again? this sentence is not clear, since the maximum height is the same as before 11:20 UTC

The sentence was modified.

  • Section 3.3:
    1. “Information regarding weather data on February 21 was obtained from the forecast system provided to INGV-OE by ARPA” ? is it really a forecast or based on actual meteorological data?

They are forecasts. We added more information about meteorological data used in this study.

  1. The model considers the ejection point, which does not necessarily corresponds to the zone of magma fragmentation. The magma can fragment somewhere else, then the fragments can be accelerated within the volcanic jet until they are ejected into the atmosphere at a different point.

As suggested by the reviewer, the fragmentation point was not considered in this study. Hence, we deleted any sentences related to the fragmentation process in order to clarify that we are considering only different ejected points.

  1. “The distance between the takeoff point and the deposition point of the clast cannot be considered constant, because the fragmentation process during lava fountains is very complex and can occur not only at the crater rim” ?Again, you are talking about the ejection process; the dynamics from fragmentation to the ejection point is very complex, and it is not taken into account in the Eject model

Done, the sentence was deleted.

  1. what is the estimated Reynolds number for the observed ballistics?

Considering the size range of the particles measured in the field (from 4 to 10 mm), the Reynolds numbers range between about 1 x  10^5 and 4.5 x 10^5. 

  1. what is the estimated Mach number for the observed ballistics?

Mach number ranges between 0.1 and 0.2

  1. “The Eject! program assumes that the drag coefficient for spherical shaped blocks is: Cd = 5 for 200 < Re < 2×105 and Cd = 0.1 for Re > 2×105. Correlating the drag coefficient with the Mach number, values considered for a sphere are: Cd = 0.5, 0.8 and 1.” ? Did you take these values as constants, or took the variable cd for sphere in Eject!?

We took a constant Cd; this was added in the text.

  1. “However, it should be considered that from these height values the topographic elevation between the ground level and the SEC must be subtracted (3350 - 2750 m = 600 m), so the actual height of the IJR considered is between 2500 and 3300 m beyond the crater rim.” ? Do you mean the bottom of the SEC? And which ground level?

Done, the sentence was modified.

  1. “Values obtained from the first set of tests are in very good agreement with the field data, and were used to carry out a second round of simulations, varying the ejection angles. In this case, trajectories allowing the clasts to reach the real covered distance are obtained using angles between 20° and 50°.” ? Where are these results? I would like to see a plot showing the horizontal range vs the ejection angle for different diameters and cd values.

Done.

  • Section 3.4:
    1. It is not clear to me whether you are talking about cm-size ballistic fragments or tephra fallout from the eruptive column.

Done, the sentence was modified.

  1. Do you have evidence that the ballistic reached these localities on the specific dates, or this is just a working hypothesis?                        

                      Yes, we have had communication on the fallout of large clasts.  

  1. “which was set at a value of 0.1 with an artillery shell type clast shape associated.” ?Why?

We have the maximum distance reached by ballistics. It was added in the paper.

  1. Values of densitis are much higher than the ones presented in Table 1!!! Why?

Yes, it  was a mistake. Those densities were not measured and they were deleted by the table.

  1. The HIJR values presented here are determined from the TIR camera or from the ballistic analysis?

Yes, by the TIR camera.

  1. “∼230 m/s, 3000 m and 40° for the February 18 episode; 180 m/s, 2000 m and 65°for the February 23 episode; 275 m/s, 3800 m and 60° for the February 28 episode; 230 m/s, 2700 m and 50° for the October 23, 2021 episode. Regarding clasts characterized by diameter of 10 cm, values of the same parameters obtained from the simulations are respectively: 200 m/s, 2800 m and 45° for the February 18 episode; 180 m/s, 2000 m and 65° for the February 23 episode; 280 m/s, 4000 m and 70°for the February 28 episode.” ? Al these numbers should be presented in a Table.

Done, a new table was added in the text.

  1. “Decimeter-sized clasts were not found for the October 23, 2021, episode near the area of Rifugio Citelli.” ?Were they found somewhere else?

The sentence was deleted.

  • Section 4 (Discussion):
    1. the analysis of the influence of the wind velocity is missing in the results section.

This analysis is lacking because we have the information obtained by weather forecasts that are widely tested since 2006 (Scollo et al. 2009).

  1. The clast diameter and density are also very important factors.

This was added in the text.

  1. Our tests make available estimation of the maximum distance reached by the clasts as these factors are changing” ? rewrite

The sentence was re-written.

  1. This could be related to the incorporation of a number of clasts as large as >5-10 cm within the eruptive column, which were transported up to the higher portions of the eruptive column, where they are most affected by the winds and can precipitate at larger distances from the point of emission.” ? Do you have evidence of such fragments?

Yes, we observed in the past. We added two references. 

  1. “this study assume that the fragmentation process during lava fountaining” ? This study is based on simulations with the eject code, which itself does not consider fragmentation process. In any case, the assumptions in the model and in this study, correspond to the ejection point.

Yes, the reviewer is right. As the fragmentation process is not considered in this study we deleted the sentence.

  1. “This is why we retain that the points from which the ballistics are distributed in the entire lava fountain height and not only above the vent.” ? rewrite

We rewrote the discussion and this sentence was deleted.

  1. “Obviously, the ejection angle should be set to 45°.” ? This is not always the optimum ejection angle. It depends on the altitude difference between the ejection and landing point, and the drag force. You could observe this if you make a plot of the horizontal range vs ejection angle for different conditions.

Yes, the sentence was deleted.

  1. “Typically, the blocks fall ballistically, making trajectories of a few kilometers, whereas lighter products such as scoriae can sometimes reach distances even greater than 10 km from the vent” ? According to the pure ballistic model, denser particles should travel farther than lighter particles, because the drag force is inversely proportional to the clast density. This observation implies that their trajectory is not purely ballistic, which is already well known in the literature. The limitations of the pure ballistic model (such as eject) should be discussed in more detail.

               We agree, a new sentence was added to clarify that those particles are not considered in the present study.

  1. “A comparative study between multiple models for the ballistic trajectories of clasts could therefore be useful in the future to assess in more detail the hazard” ? such as?

A new paper concerning the trajectory of ballistics has been added.

Equation (1): one expression for this equation is enough, solved either for vex or HIJR, but not for both, or at least leave some space between both expressions. It is confusing in its current form

Yes, it was a typo. The formula was corrected.

Figures:

  • Figure 4 and 7: A vertical scale or reference is needed

They are images from cameras. Adding information was added in the text.

  • Figure 5(a): these velocities are derived from Eq. 1? If so, you should say it

Done, we improved the caption of Figure 5.

  • Figure 6: Instead of presenting this captures of the eject interphase, I suggest making a four panel figure with your own plots showing in each plot the trajectories with the a different diameter and the three Cd values considered, using a more appropiate scaling to see the differences. These captures are not needed, since you already provide in Table 2 all the input and output parameters and the plots show different runs which are not specified.

Done.

  • Figure 9: Most of these event have lower lava fountain heights and lower ejection velocities than the the February 21, 2022 event. Again it is not clear to me whereas you did observe ballistcs at several km from the source, or if you are considering a worst-case scenario.

See previous comments.

Tables:

  • Table 1: For the purpose of the model, it is more appropiate to take the geometric average of the perpendicular diameters, i.e. the cube root of the product of these diameters, since it represents the diameter of a sphere with the same volume

Done, it was added to the table.

Round 2

Reviewer 1 Report

I have reviewed the changes made in this manuscript since the last submittal.  Several aspects of the paper have improved substantially. The authors provided line numbers as requested, and have greatly improved the English.  They have also clarified some passages that were unclear in the past version, have added symbols to Fig. 1b clarifying features on their map, and more clearly explained the various lines in plots in Fig. 7 that were unclear previously. 

Some other issues are still problematic:

·         Several points specified below are still not explained clearly.  Most important of these is the elevation of the takeoff point used in these simulations.  Lines 258-259 say “different ejection heights here were considered between the elevation of the crater rim of the SEC and the maximum HIJR”.  Does “ejection heights” mean “takeoff point elevations”?  This should be stated more clearly, and there should be some explanation of why such a range of heights was chosen. 

·         The clasts analyzed are 4-10 cm in diameter, which is much smaller than those typically considered ballistic blocks.  This was a key reason why I recommended rejection in the last review.  These small fragments will be influenced by jet updrafts, and interactions with other blocks.  The authors have added some discussion of these processes in the Discussion section, which partially alleviates the problem.  However, they should acknowledge that they are using unusually small blocks, subject to other influences, near line 268 when they first discuss block sizes. 

·         During their field measurements, the authors do not say how they know that the blocks they examined were actually ejected during the most recent eruptive events.

These problems are significant but fixable.  Once they have addressed these, and the additional technical points below, I think that the paper will be worthy of publication.

Larry Mastin

Specific comments:

The revised manuscript contains a great deal of yellow highlighted text.  The highlights can’t be removed as far as I can tell, making the manuscript difficult to read and edit.  The authors should have cleaned up this manuscript before submitting.  And the Editor should have required the authors to re-submit a clean copy before sending it out to review.

Line 14: change “fall out” to “fallout”

Lines 44-45:  change “as high as up to 15 km” to “as high as 15 km”

Lines 85-87:  when you did your field sampling, how did you know that the blocks that you measured were ejected during the most recent eruptive episode?

Line 94:  why do you use b/a rather than c/a to get aspect ratio?

Lines 103-104:  this sentence is incoherent.  What exactly has been named the incandescent region?

Line 133:  change “in the unit of time” to “per unit time”

Lines 147-148:  change “escape velocity” to “ejection velocity”

Line 148:  change “in degrees” to “in degrees from horizontal”

Line 187:  change “ballistivs” to “ballistics”

Line 217:  change “weight” to “mass”, and consider changing the symbol from “W” to “M”.

Line 233: change “photograms” to “photographs”

Line 258:  does “ejection heights” mean “elevation of takeoff point elevations”?  It’s important to be clear.  If you do mean that you specified the takeoff-point elevation to be in the middle of the incandescent jet somewhere, some explanation for this choice is required.

Figure 5:  Are these the images you used to obtain IJR height?  I don’t see a height scale on them.

Line 288: change “worth to note” to “worth noting”

Line 291:  change “other four” to “four other”

Line 322: delete “and” from before “considered”

Figures 10 and 11:  these figures have been switched.

Lines 339-345:  These sentences are not clear.  It looks like you’re trying to describe the sequence of simulations you used in order to constrain the combination of inputs that would send the ballistic blocks to the observed distances.  Is that right?  I don’t follow your description however.

Line 371:  change “systems is possible” to “systems, it is possible”

Lines 376-393:  since you are analyzing clasts that are generally smaller than those considered ballistics, it is likely that these clasts were subject to strong air drag, but that the air drag could have been greatly reduced if they were flying behind other, larger clasts.  This reduced air drag could explain why some small clasts went farther than expected.

Line 409: change “calibrated” to “used”

Author Response

Reviewer 1:

I have reviewed the changes made in this manuscript since the last submittal.  Several aspects of the paper have improved substantially. The authors provided line numbers as requested, and have greatly improved the English.  They have also clarified some passages that were unclear in the past version, have added symbols to Fig. 1b clarifying features on their map, and more clearly explained the various lines in plots in Fig. 7 that were unclear previously.

Some other issues are still problematic:

  • Several points specified below are still not explained clearly. Most important of these is the elevation of the takeoff point used in these simulations.  Lines 258-259 say “different ejection heights here were considered between the elevation of the crater rim of the SEC and the maximum HIJR”.  Does “ejection heights” mean “takeoff point elevations”?  This should be stated more clearly, and there should be some explanation of why such a range of heights was chosen.

 Yes. Thanks. We replaced it with the takeoff point elevations. Moreover, we added a new sentence to clarify it.

The clasts analyzed are 4-10 cm in diameter, which is much smaller than those typically considered ballistic blocks.  This was a key reason why I recommended rejection in the last review.  These small fragments will be influenced by jet updrafts, and interactions with other blocks.  The authors have added some discussion of these processes in the Discussion section, which partially alleviates the problem. 

We are very sorry. We clarify the size of clasts in the field. Most of the clasts around the SEC had a size greater than 64 mm, although large lapilli were also found. We clarify this point by focusing our simulations only on the blocks greater than 64 mm.  

However, they should acknowledge that they are using unusually small blocks, subject to other influences, near line 268 when they first discuss block sizes.  

We thank the reviewer for this suggestion. We added a new sentence when we introduced the Eject! software. Moreover, in order to avoid misunderstanding we focus our simulation only on blocks ( not large lapilli). Consequently, the clast size of 4 cm was not more considered in the simulations.

During their field measurements, the authors do not say how they know that the blocks they examined were actually ejected during the most recent eruptive events.

One of the authors went on the field before and after those eruptive events as a volcanologist on duty. Moreover, we also had some additional information by volcanological guides who  accompany thousands of tourists throughout your excursions in the area analyzed in the field. 

These problems are significant but fixable.  Once they have addressed these, and the additional technical points below, I think that the paper will be worthy of publication.

Larry Mastin

We really thank Dr. Larry Mastin and we hope that most of the weak points of this work were clarified.

The revised manuscript contains a great deal of yellow highlighted text.  The highlights can’t be removed as far as I can tell, making the manuscript difficult to read and edit.  The authors should have cleaned up this manuscript before submitting.  And the Editor should have required the authors to re-submit a clean copy before sending it out to review.

We are very sorry. There was a clean up version of the manuscript but we did not see a way to add it in the review system.

  • Line 14: change “fall out” to “fallout”

Done.

  • Lines 44-45:  change “as high as up to 15 km” to “as high as 15 km”

Done.

  • Lines 85-87:  when you did your field sampling, how did you know that the blocks that you measured were ejected during the most recent eruptive episode?

One of the authors went on the field before and after those eruptive events as a volcanologist on duty. Moreover, we also had some additional information by volcanological guides who accompany thousands of tourists throughout your excursions in the area analyzed in the field. A brief sentence was added in text.

  • Line 94:  why do you use b/a rather than c/a to get aspect ratio?

We modified the aspect ratio definition and we replaced the values in the table.

  • Lines 103-104:  this sentence is incoherent.  What exactly has been named the incandescent region?

This sentence has been re-written.

  • Line 133:  change “in the unit of time” to “per unit time”

Done.

  • Lines 147-148:  change “escape velocity” to “ejection velocity”

Done.

  • Line 148:  change “in degrees” to “in degrees from horizontal”

Done.

  • Line 187:  change “ballistivs” to “ballistics”

Done.

  • Line 217:  change “weight” to “mass”, and consider changing the symbol from “W” to “M”.

Done.

  • Line 233: change “photograms” to “photographs”

Done.

  • Line 258:  does “ejection heights” mean “elevation of takeoff point elevations”?  It’s important to be clear.  If you do mean that you specified the takeoff-point elevation to be in the middle of the incandescent jet somewhere, some explanation for this choice is required.

The sentence was modified.

  • Figure 5:  Are these the images you used to obtain IJR height?  I don’t see a height scale on them.

Yes. It is right. The calibration is a further analysis described in Mereu et al. (2020) and follows a similar procedure described in Sciotto et al. (2019) https://www.nature.com/articles/s41598-019-52576-w

  • Line 288: change “worth to note” to “worth noting”

Done.

  • Line 291:  change “other four” to “four other”

Done.

  • Line 322: delete “and” from before “considered”

Done.

  • Figures 10 and 11:  these figures have been switched.

Done.

  • Lines 339-345:  These sentences are not clear.  It looks like you’re trying to describe the sequence of simulations you used in order to constrain the combination of inputs that would send the ballistic blocks to the observed distances.  Is that right?  I don’t follow your description however.

The reviewer is right. This sentence was modified and Figure 10 was deleted because our results are described in Table 3.

  • Line 371:  change “systems is possible” to “systems, it is possible”

Done.

  • Lines 376-393:  since you are analyzing clasts that are generally smaller than those considered ballistics, it is likely that these clasts were subject to strong air drag, but that the air drag could have been greatly reduced if they were flying behind other, larger clasts.  This reduced air drag could explain why some small clasts went farther than expected.

This sentence was added in the text.

  • Line 409: change “calibrated” to “used”

Done

Reviewer 3 Report

I appreciate the effort made by the authors to revise the paper. The manuscript has been greatly improved and all my major concerns have been addressed. In the attached PDF file there are some final minor corrections. Once corrected, I think the manuscript would be ready to be published in Geoscience.

Author Response

Reviewer 2:

  • Line 21: Eject! is not a model, it is a software based on a simple ballistic model used previously by several authors.

Done

  • Line 52: change “source of hazard” to “threat”.

Done

  • Line 89: add “in diameter”

Done

  • Line 90: “We measure their weight with a digital balance (in grams, g)” ? or in kg, to make units consitent with the density.

Done.

  • Line 141: change “model” to “software”.

Done

  • Line 195: change “5” to “4”, and change “15” to “12”.

Done

  • 1: add “(m)” below dg.

Done

  • Line 272: add “respectively”.

Done

  • Figure 7: the colors represent different angles, right? please indicate it in the caption and check the numbers in Figure 7 (d).

Done

  • Line 286: change “or” to “and”.

Done

  • Line 288: change “drive” to “control”.

Done

  • Line 341: change “show that the” to “with the defined”.

Done

  • Line 364: change “size” to “density”.

Done

  • Line 369: change “is possible” to “can be used”.

Done

  • Line 379-380: “Furthermore, considering the maximum speed of IJR, allow us to identify a larger area that could be affected by the fallout of ballistics avoiding any threat to the population”: this sentence is not clear.

Done

  • Line 397: add “, thus”.

Done

  • Line 398: add “and for this reason”.

Done